# Trends and Missing Links in (De)Hydration Research: A Narrative Review

**DOI:** 10.3390/nu16111709

**Published:** 2024-05-30

**Authors:** Alexandre Rebelo-Marques, Bruna Coelho-Ribeiro, Adriana De Sousa Lages, Renato Andrade, José Afonso, Rogério Pereira, Ana Sofia Batista, Vitor Hugo Teixeira, Cristina Jácome

**Affiliations:** 1Faculty of Medicine, University of Porto, 4200-450 Porto, Portugal; 2Coimbra Institute for Clinical and Biomedical Research (iCBR), Faculty of Medicine, University of Coimbra, 3000-370 Coimbra, Portugal; 3Magismed Innovation Institute, 4710-353 Braga, Portugal; 4Life and Health Sciences Research Institute (ICVS), School of Medicine, University of Minho, 4710-057 Braga, Portugal; 5ICVS/3B’s-PT Government Associate Laboratory, 4806-909 Guimarães, Portugal; 6Endocrinology Department, Hospital de Braga, 4710-243 Braga, Portugal; 7Clínica Espregueira—FIFA Medical Centre of Excellence, 4350-415 Porto, Portugal; 8Dom Henrique Research Centre, 4350-415 Porto, Portugal; 9Porto Biomechanics Laboratory (LABIOMEP), Faculty of Sports, University of Porto, 4200-450 Porto, Portugal; 10Centre of Research, Education, Innovation, and Intervention in Sport (CIFI2D), Faculty of Sport, University of Porto, 4200-450 Porto, Portugal; 11Higher School of Health Fernando Pessoa, 4200-253 Porto, Portugal; 12Faculty of Nutrition and Food Sciences, University of Porto, 4150-180 Porto, Portugal; 13Research Center in Physical Activity, Health and Leisure, CIAFEL, Faculty of Sports, University of Porto, FADEUP, 4200-540 Porto, Portugal; 14Laboratory for Integrative and Translational Research in Population Health, ITR, 4050-600 Porto, Portugal; 15CINTESIS@RISE, MEDCIDS, Faculty of Medicine, University of Porto, 4099-002 Porto, Portugal

**Keywords:** hydration, dehydration, health, performance, athlete, sports, sports drinks, water, research gaps, fluid intake

## Abstract

Despite decades of literature on (de)hydration in healthy individuals, many unanswered questions remain. To outline research and policy priorities, it is fundamental to recognize the literature trends on (de)hydration and identify current research gaps, which herein we aimed to pinpoint. From a representative sample of 180 (de)hydration studies with 4350 individuals, we found that research is mainly limited to small-scale laboratory-based sample sizes, with high variability in demographics (sex, age, and level of competition); to non-ecological (highly simulated and controlled) conditions; and with a focus on recreationally active male adults (e.g., Tier 1, non-athletes). The laboratory-simulated environments are limiting factors underpinning the need to better translate scientific research into field studies. Although, consistently, dehydration is defined as the loss of 2% of body weight, the hydration status is estimated using a very heterogeneous range of parameters. Water is the most researched hydration fluid, followed by alcoholic beverages with added carbohydrates (CHO). The current research still overlooks beverages supplemented with proteins, amino acids (AA), and glycerol. Future research should invest more effort in “real-world” studies with larger and more heterogeneous cohorts, exploring the entire available spectrum of fluids while addressing hydration outcomes more harmoniously.

## 1. Introduction

Since the first species ventured from the oceans to live on land, water has been a significant key to survival, representing one of the most essential needs for all forms of life [1]. Indeed, the proper function of various vital systems, such as the cardiovascular, respiratory, kidney, liver, brain, and peripheral nervous system, depends on an adequate hydration status [2,3]. Hydration status is defined as the balance between water inputs (the water we drink, the water we “eat” and the water we produce) and outputs (excreted mainly by the kidneys, skin, and respiratory tract) [2,4,5]. Over 24 h, such a balance must equal zero to keep total body water (TBW) constant [2]. To guarantee fluid balance, a network of interconnected and homeostatic mechanisms comes into action to keep plasma osmolality (Posm) within narrow limits [1,2,6,7,8]. Neuronal sensors, the renin–angiotensin system, arginine vasopressin (AVP), antidiuretic hormone (ADH), and thirst are part of the big picture [2,6,7,8,9].

Nonetheless, an imbalance between water intake and losses may still occur, potentiating dehydration [10]. Dehydration is described as a 1% or more significant body weight loss due to fluid loss [11]. Between 1% and 4%, fluid losses lead to progressive declines in athletic performance, thermoregulation, and appetite [2,11]. Therefore, a clear interconnectivity is established: water, hydration, and health are three inseparable dimensions.

Prolonged exercise leads to the loss of body fluid related to elevated sweat [12]. Thus, during sports competitions, water reaches its utmost importance, as maintaining fluid homeostasis is fundamental to optimizing athletic performance [13]. Body temperatures (core and skin) will significantly increase during exercise, depending on the environmental conditions (temperature, humidity, altitude, wind exposure, etc.) [14]. Then, the excess heat produced by the body temperature elevation or absorbed from the environment is dissipated through sweat evaporation [14,15]. Considering that sweat is typically hypotonic, exercise-induced sweat losses lead to hypertonic hypovolemia, contributing to an increment in cardiovascular and thermoregulatory strain, an elevated perception of effort, and a decrement in muscle blood flow and aerobic reserve, especially in hot conditions [13,15,16,17]. In parallel, body water and electrolyte stores are reduced, which must be counterbalanced by consuming fluids during the exercise or, to a lesser contribution, by hyperhydration hours before the event [15,17,18]. The problem arises from the incapacity of high-performance athletes to have the opportunity or desire to drink at a rate to keep pace with sweat losses [15]. These athletes incur a body fluid deficit where dehydration (loss of ≥2% of body mass) and hyponatremia (blood sodium < 135 mEq/L) are achieved, adversely impacting athletic performance and potentially their health, mainly when core and skin temperatures are concomitantly high [14,15,19]. Such a fluid deficit is exacerbated when athletes start exercising in a hypohydrated state [13]. Conversely, hyperhydration may also be problematic during prolonged exercise by increasing the risk of hyponatremia [19,20,21].

Despite the more than 50 years of literature on hydration for sporting events, many questions still need to be answered. Guidelines for fluid intake have distinctly evolved over the past decades [14,22,23], but there still needs to be an evidence-based and standardized hydration strategy [15]. Indeed, current evidence suggests two opposite approaches: programmed/planned drinking vs. drinking to thirst/ad libitum drinking [15]. Furthermore, there is still no consensus on whether water is the ultimate fluid for hydration or if consuming beverages containing electrolytes and carbohydrates (CHO) can add several benefits over water alone [14,21]. So, what we must learn about (de)hydration appears significant. Also, the extent to which dehydration impairs athletic performance remains a critical question. Likewise, there is yet to be a consensus on which fluid and amount we should drink. The real question is what we know.

This narrative review aims to clarify the research trends and what is known about hydration in healthy individuals. This can advise future priorities on research, policy, and funding. We selected interventional or observational studies that evaluated the impact of different hydration or dehydration strategies on healthy individuals to summarize current trends on protocols and analysis of (de)hydration. Our representative literature sample comprises 4350 individuals from 180 studies [22,23,24,25,26,27,28,29,30,31,32,33,34,35,36,37,38,39,40,41,42,43,44,45,46,47,48,49,50,51,52,53,54,55,56,57,58,59,60,61,62,63,64,65,66,67,68,69,70,71,72,73,74,75,76,77,78,79,80,81,82,83,84,85,86,87,88,89,90,91,92,93,94,95,96,97,98,99,100,101,102,103,104,105,106,107,108,109,110,111,112,113,114,115,116,117,118,119,120,121,122,123,124,125,126,127,128,129,130,131,132,133,134,135,136,137,138,139,140,141,142,143,144,145,146,147,148,149,150,151,152,153,154,155,156,157,158,159,160,161,162,163,164,165,166,167,168,169,170,171,172,173,174,175,176,177,178,179,180,181,182,183,184,185,186,187,188,189,190,191,192,193,194,195,196,197,198,199,200]. The methods of selecting the representative sample of the literature are described in Appendix A. Below, we present a narrative summary and discussion of each topic concerning (de)hydration based on the representative sample of the literature.

## 2. What Are the Current Trends and Gaps in Study Implementation?

Studies started being published in 1986, with a considerable chronological publishing increase since 1990 (Figure 1A). Within our representative sample (studies from 1986 until 2023), 16 studies have been published since 2021, corresponding to 20% of the papers published in the last decade showing an increasing interest in (de)hydration topic.

Studies were dispersed across 27 countries, primarily from North America (52%) and Europe (24%), and a minority identified from Oceania, South America, Central America, and Africa (Figure 1B,C). The literature unveils a geographic bias toward high-income countries, with investigation from low- or middle-income countries usually underrepresented [201]. The same trend seems to be prevailing in the hydration topic. Among various causes, genetic, epigenetic, and environmental factors prompt apparent differences in the human population [202]. Such global variability has been a long-standing reality worldwide [202] that turns the geographical bias observed in this review into a preoccupant gap of the hydration theme.

Most of the research comprises interventional studies (number of articles (k) = 171, 95%), mostly randomized (70%; Figure 1D), comparing different (de)hydration strategies (Figure 1E). This finding is relevant and encouraging because randomized interventional studies are the most fit to investigate (de)hydration strategies.

## 3. Are Sample Sizes Powered Enough to Draw Meaningful Conclusions?

A total of 4350 participants were involved across the included studies, with an average of ~24 participants per study (median: 12), ranging from 5 to 573 individuals. Most studies (65%) comprised small-scale sample sizes, including less than 15 individuals. Nevertheless, evaluating hydration in a large sample is particularly interesting to improve preventive and intervention strategies. This could be achieved by conducting epidemiological and extensive clinical studies [5]. Despite not having a particular number of subjects to integrate a cohort, the minimum sample size should be 10–20 times the independent and adjusted continuous variables of the study [203]. The studies included in this review mainly cover at least two independent outcome variables, demonstrating that the cohorts investigated are characterized by a reduced sample size, which may jeopardize the statistical power and external validity of the studies. In summary, most knowledge on hydration is limited to underpowered studies with small sample sizes, which is a common limitation across sports research [204,205,206,207].

## 4. Who Is Being Studied, and Who Is Left Behind?

There is a disparity in the research in males, and sex differences in hydration research have already been documented [13]. In the representative sample of the 180 studies reported here, there were 2497 males (57%) and 943 females (22%). A total of 114 studies (63%) studied exclusively males, while six studies (3%) were exclusively females (Figure 2A). Even in mixed-sample studies (including both females and males; 27%), males predominated with a ratio of 1.2 males/females. The biological differences between sexes are essential in many research areas, including (de)hydration. For example, when fluid consumption relative to body mass is similar between males and females, males have more significant sweat losses [13]. Research on collegiate athletes showed that males consistently display higher degrees of dehydration than their female counterparts [13]. The overrepresentation of males in the hydration literature underpins that the existent global information regarding hydration skews toward the cumulative effect of sex variability. Considering the considerable biological variability of females because of their complex hormonal system [56,57,58,59,60], it is preferable to use a female cohort rather than a male one, thus guaranteeing that differences in results are after the study intervention rather than the inner cohort variability. As a result, females are typically underrepresented in the sports sciences investigation [208,209,210,211,212]. However, this belief related to biological variability has already been deconstructed by Souza et al. [213], which stated that menstrual status (eumenorrheic vs. amenorrheic) does not influence exercise performance in female athletes. There is higher interindividual variation in energy expenditure (defined as the metabolic costs of multiple physical, physiological, and behavioral traits) among males. With aging, this high variation in total energy expenditure decreases in males [214]. The latter may account for substantial economic and social burdens as well as for outcomes compromised by the premise of higher variation in the total energy expenditure of males.

Age research gaps can also play an essential role in skewing research findings. Most of the studies (86%) included subjects with a mean age ≥18 years (Figure 2B), but the average age was 22.4 years, indicating that research mainly studies young individuals. This finding aligns with most research on healthy and physically active individuals [52], where young adults are naturally the most studied population. Only 8 studies investigated adolescents (11%) or children (3%), and older adults (≥65 years) were utterly overlooked, only being included in 1 study [215]. Considering that young children and the elderly incur a higher risk of hydration disturbances [5,216,217], the existent literature is characterized by a preoccupant non-homogenous age dispersion, with a prevalent health problem being under-recognized and poorly managed.

Research on (de)hydration is essential to investigate its impact on health but is also especially relevant for sports. Indeed, around half of the studies (55%) included individuals characterized as athletes (i.e., at least Tier 2 of the Participation Classification Framework (PCF) (218), and 29% recreationally active individuals (Tier 1) (Figure 2C). Research on athletes was most common in Tier 2 (21.7%), followed by Tier 3 (15%) and Tier 4 (6%). Interestingly, no study exclusively included world-class athletes (Tier 5). The absence of studies in Tier 5 may be explained by the difficulty in enrolling high-level athletes in research studies due to their training schedules [207,218]. Thus, they are often a minority in research. This underlying problem results in recreationally active individuals being the most studied, despite subjects of the integrated cohorts mainly being referred to as athletes. Thus, an unveiling trend exists to primarily investigate (de)hydration in healthy subjects instead of athletic individuals. Nonetheless, when athletes are the focal point, local-level representation (Tier 2) is the most considered tier, with a narrow focus on high-competition athletes (Tiers 3 and 4) or even leaving out world-class athletes (Tier 5). Despite hydration playing a pivotal role in athletic performance [11,13], it appears to be highly neglected in this population, representing a vital literature gap that must urgently be addressed.

## 5. Are We Outlining the Most Relevant Study Strategy or Getting Stuck into a Loop?

The intervention studies used a primary strategy that broadly fit into two main branches: hydration and dehydration. Studies with a hydration intervention were divided into 4 key categories: (1) ingestion of different types of beverages (64%); (2) ingestion of similar beverages with different nutritional compositions (32%); (3) ingestion of similar beverages at different temperatures (5%); and (4) ingestion of fluid at different intake kinetics (2%) (Figure 1E). Observational studies (k = 9, 5%) were classified into a further category, where the main goal concerned descriptive data regarding reference values for hydration status, beverages and sodium intake patterns, urine output, perceived fluid consumption, and physical activity of healthy athletes or the general population.

Ingestion of different types of beverages (64%) was by far the most studied strategy, with 61 studies (36%) comparing 2 distinct beverages and 5 studies (3%) analyzing 1 fluid. Such numbers denote an evident trend in exploring solely the effect of ingesting various beverages on hydration status. Currently, the literature neglects nutritional composition, temperature, and intake kinetics, which are variables that have the potential to change the hydration status panorama completely. How can we genuinely grasp hydration if we do not contemplate all its pillars? It is clear that current research is digging over and over the same topic, needing to switch its direction. Herein, we discuss the everyday trends and gaps in strategies for (de)hydration of available research.

## 6. Which Exercise Protocol Stimulates Dehydration? Are They All the Same?

Most studies (89%) applied an exercise protocol with more than 18 different exercise types. Cycling (48%), running (17%), treadmill (9%), and football/soccer (8%) were the most represented exercises, and each of all other types of exercise was denoted in less than 8% of studies (Figure 3A). The tendency of exercise protocols towards cycling and running is evident, which may be related to the fact that cycling and running/treadmill are exercises whose variables, such as elicited heart rate, are feasibly more controlled indoors/laboratory. Looking at the big picture, it is quickly fathomable that the hydration area is primarily confined to recreated laboratory environments that cannot replicate the motivational aspects of real-life competitions, where extra effort might be elicited [15,16,17]. Valid ecological conditions are critical in determining how the physiological response is affected by dehydration in athletic performance [16]. Hence, it is imperative to translate scientific research into practical, real-world settings to grasp this topic. The solution goes by extending the horizons to match the panoply of available sports, especially the most practiced ones (e.g., football, cricket, hockey, tennis, and volleyball [219]) and those whose performance has a more significant link with (de)hydration (performed under hot environments and in long-lasting exercise [220]).

The exercise protocol was either to prompt dehydration or assess hydration during exercise. Studies that included an aerobic exercise protocol (77%) averaged 112.6 min (mode 90 min). Studies that applied a resistance exercise or a walking protocol were characterized by an average of 5.5 rounds of the same exercise (sets, 6%) and 44 repetitions within the same round (reps, 4%) or 31.7 km (km, 17%). Participants were exposed to exercise until volitional exhaustion (inability to continue exercising) in 20% of studies (Table 1). Duration of the exercise protocol was inconsistent and unclear across studies, with commonly unreported parameters (frequency, intensity, time, type, and volume of exercise), precluding the inclusion of all studies into the analysis. Despite protocol variability being inherent to any research protocol across any area, there is also a need for reproducibility and replicability, which needs to be improved in hydration studies. The percentage of body weight lost was used as the endpoint of the protocol in 16% of studies, where dehydration was frequently defined as the loss of 2% of body weight. This aligns with the American College of Sports Medicine Guidelines on Nutrition and Athletic Performance [72], which defines fluid deficits of >2% body weight as detrimental to cognitive and exercise performance. Nonetheless, the maximum body weight loss value registered across the 23 studies was 4%, representing double the reference value [221]. Could the subjects be suffering from an excess of dehydration? Are we replicating real-world parameters? Again, the differences between laboratory-simulated vs. real-life environments emerge.

## 7. Temperature and Humidity: Do They Play a Role?

Environmental conditions, such as temperature, humidity, and air velocity significantly impact athletic performance and hydration status [220,222]. For instance, water lost through respiration may be estimated based on ventilatory volume and the relative humidity of the environment, being promoted by low, close humidity values [222] when exercise is combined with elevated ambient temperatures, the core temperature of the subject substantial rises, compromising endurance capacity [220]. On the other hand, in temperate conditions, dehydration by 1–2% of body mass has no impact on endurance exercise performance when the exercise duration is no longer extended than 90 min [220]. Observing the environmental conditions to which the subjects are exposed during the integrated study interventions becomes critical.

Ambient conditions should have been addressed, with 28% and 39% of studies not reporting temperature and relative humidity, respectively. When reported, normal environmental conditions were the most frequent, with 39% and 33% of the studies registering average temperature (15–30 °C) and relative humidity (30–60%). Hot environments (temperature above 30 °C) [223] (25%) with relative humidity above normal (above 60%) [224] (13%) were less frequent (Figure 3C). Research on hydration does not consider the impact that ambient conditions may have on hydration status and, consequently, on exercise performance. Future studies should give more importance to these environmental features and investigate subjects in more adverse environmental conditions.

## 8. Which Beverages and Nutritional Components Are in the Spotlight?

An evident key to avoiding the harmful impact of fluid loss on exercise performance is to replace fluid deficit through oral consumption [175]. The interest in the type and composition of beverage consumption is familiar. Still, there is no global validation concerning the specific type of rehydrating fluid since nutrition goals and requirements are not static [175,221]. Approximately half of the studies (51%) reported water as the ingested fluid. Such a pattern is expected since the most accessible and standard form to replace fluid loss is to drink plain water [225]. However, especially for beverages intended to improve physical performance, optimal fluid replenishment is often formulated with other ingredients, such as CHO and/or electrolytes, to enhance palatability, stimulate thirst, speed intestinal fluid absorption, and prompt fluid retention [225]. Congruently, CHO (31%), CHO-E (21%), and sports drinks, besides Gatorade and Powerade (20%), were other commonly investigated beverages (Figure 3D). Different varieties englobed less common beverages, such as honey, coconut water, maple water, and juices.

The optimal composition may depend upon the source of the fluid loss (sweat, urine, respiration, or diarrhea/vomiting), the objective of the fluid intake, the target population, and the environmental conditions [225]. The primary purpose of a beverage is to replace water losses, but it may also serve as a vehicle for supplying nutrients to strengthen physiological function. For example, CHO impacts the water absorption rate, sweetness, and palatability of beverages, and it is essential for maintaining blood glucose and high rates of CHO oxidation, especially during physical activity [225]. Accordingly, CHO was the most stated element (66%), and sugars were claimed in 31% of the included studies, representing nearly half (46%) of the reported CHO (Figure 3E).

Similarly, electrolytes augment palatability and stimulate the physiological drive to drink, promoting renal water reabsorption, which aids in maintaining extracellular fluid volume (in the case of sodium) and affects neural transmission, muscle contraction, and vascular tone (resulting from the balance of potassium in the extracellular versus intracellular space) [21,221]. Mainly, when prolonged exercise is coupled with aggressive fluid replacement, it is necessary to prevent hyponatremia (blood sodium < 135 mmol/L, also known as water intoxication) and even a remarkable risk of muscle cramping, which may impair athletic performance [221,226]. In this context, following CHO, sodium (58%) and potassium (41%) are the most preferred compounds (Figure 3E), with sodium being the most mentioned electrolyte. Chloride is also usually found in beverages as it is the major anion lost in sweat [225,227]. Nonetheless, chloride appears to be a neglected electrolyte, with only 14% of the studies mentioning it.

Protein is included in postexercise fluids to promote muscle protein synthesis, helping the recovery process following exercise [225]. Protein intake may also play a role in rehydration. Milk has been suggested to enhance fluid retention during the post-exercise rehydration process [225]. Only 8% and 6% of studies utilized beverages supplemented with proteins (CHO-PRO) or milk.

Drinks supplemented with amino acids (AAs) were the least common (CHO-AA, 1.7%; CHO-E-AA, 0.6%; E-AA 0.6%), as well as the compound AAs (4%; Figure 3D,E). The small intestine has membrane transporters relying on ingested AAs. The AAs, such as glutamine and alanine, appear to increase electrolyte/water absorption after exercise [225,227]. Supplementarily, branched-chain amino acids (BCAAs) have been proposed as a nutritional countermeasure of central fatigue [225]. The secondary literature advocates that substituting AAs for CHO has hinted as a potential advantage because of the carrier-mediated AA transport for intestinal absorption of sodium and water but with no added calories from sugars in the drink [227]. Studies highly reported the element CHO (66%), while drinks supplemented with AAs are the least common (CHO-AA, 2%; CHO-E-AA, 0.6%; E-AA, 0.6%) as well as the element AAs (4%).

Scientific evidence supports the efficacy of glycerol for inducing renal water absorption since it promotes a temporary hyperhydrated state when ingested with an additional volume of water [225]. Nonetheless, only 2% of the included studies use beverages with glycerol in their hydration strategy (Figure 3D). Some studies (31%) incorporate secondary characteristics, such as fat, taurine, caffeine, and vitamins.

Current research on hydration covers a wide range of beverages and their compounds. Nonetheless, evidence on beverages supplemented with AAs, protein, and glycerol must still be more manageable. Considering their importance on hydration status, fostering a more significant proportion of investigation in those less explored domains is crucial.

## 9. Does the Amount and the Way We Drink Matter?

A total of 6 different intake strategies were distinguished, with concrete fluid volume (51%) and ad libitum (25%) being the most reported. Studies averaged 2671.9 mL of total fluid ingested (mode: 1000 mL). When intake strategy was based on the percentage of body weight lost (15%), studies averaged an amount of fluid corresponding to 112% (range: 50–150%) of body weight lost, with 100% (k = 13, 62% of the studies using body weight loss as parameter for fluid intake) being the most common amount (Table 2).

Hypohydration can impair athletic performance [11,13]. However, contrary to common belief, hyperhydration can cause electrolyte imbalance, such as hyponatremia, also leading to a decline in performance or even life-threatening or fatal complications (when plasma sodium ≤ 120 mEq/L) [221,228]. In fact, over the past two decades, there has been a rising awareness that some recreational athletes drink at rates that surpass their sweat losses, promoting a state of overhydration. In such a way, it is crucial to contemplate the amount of fluid consumed pre-, during, and after exercise. Such a hydration plan must be personalized to the individual athlete field’s performance goals and practical challenges [221]. Despite the latter, the American College of Sports Medicine Guidelines on Nutrition and Athletic Performance [175] recommend a typical amount of fluid intake based on the conviction that fluid deficits of >2% body weight can compromise cognitive function and aerobic exercise performance, especially in hot environments. In cool weather, athletic performance is frequently more impacted when losing 3–5% of body weight. Specifically, two to four hours before exercise, athletes may accomplish euhydration by consuming a fluid volume equivalent to 5–10 mL/lb (11–22 mL/kg) body weight [175].

During exercise, sweat rates vary from 0.3–2.4 L/h depending on exercise intensity, duration, fitness, and environmental conditions (heat, humidity, among others) [175]. Therefore, athletes should drink the amount of fluid necessary to replace sweat so the total body fluid deficit is limited to <2% of body weight. Precisely, the fluid plan that suits most athletes will typically achieve an intake of 0.4 to 0.8 L/h [175]. After exercise, considering that athletes usually end up exercising with a fluid deficit, it is essential to restore euhydration. Adequate rehydration must consist of a volume intake more significant than the final fluid deficit, namely 125–150% of the body weight lost (1.25–1.5 L of fluid for every 1 kg of body weight lost) [175]. Ad libitum was the second most used intake strategy, raising the hypothesis that athletes may be experiencing hyperhydration. From the studies reporting an intake strategy based on the percentage of body weight (15%), the number of fluids averages 100% of body weight lost. This strategy guarantees that the total body fluid deficit does not overcome 2% of body weight. Of note, similarly to the duration of exercise protocol, information related to intake strategy could have been reported more consistently and needed clarification. Therefore, not all selected studies were eligible to be included in the analysis of intake strategy, putting again terms such as reproducibility and replicability under questioning.

## 10. Within-Session vs. In-Season Timing—Context Matters!

Hydration strategy varies depending on the timing of administration. Still, the period of the season timing may also impact the fitness status of the athletes at the time of the trial [218,229]. When within the session, both the inter- (34%) and post-set (22%) were commonly used (Figure 3F). From the combined categories, studies investigating the effects of combined pre- and inter-set (k = 30, 19%) hydration were the most frequent (Figure 3F). Considering that pre-hydration may be fundamental to prevent hypohydration during athletic performance [15,17], the observed negligence of a pre-set intervention (9%) reveals a concerning gap in the existing literature. Nonetheless, evidence also supports that fluid ingestion during exercise is desirable compared with drinking before or after sports performance [21]. This paradox puts the existing literature at an impasse. After all, are we focusing on the relevant conditions or just ignoring essential details?

Studies were characterized mainly by within-season timing (55%), which unveiled a hiatus in reporting the within-season timing, with 86% missing or insufficient information (Figure 3G). Only 14 studies offer information related to within-season timing. The competitive season (7%) is the most frequent in those, which appears like other research training approaches [230]. Further research must take within-season timing more into account.

## 11. Are We Evaluating the Most Relevant (De)Hydration-Related Outcomes?

No single measure is universally accepted to determine whether an individual is euhydrated, hyperhydrated, or hypohydrated [231]. Nonetheless, describing a trend on how hydration status is evaluated is feasible. Performance and health outcomes were also commonly studied (Figure 4).

A total of 140 studies (80%) reported a wide array of metrics on hydration status, totaling 45 distinct outcome hydration-based measures (Figure 4D). From these, 8 hydration-related metrics stand out as the most predominantly investigated: body weight changes (70%), urine outputs (e.g., volume, mass, flow rate) (49%), plasma volume changes (42%), fluid intake (40%), sweat outputs (e.g., volume, flow rate) (37%), plasma osmolality (31%), plasma electrolytes (such as Na+, K+, Cl−) (30%), and urine specific gravity (29%). Each of all other hydration-related outcome measures was assessed in less than 19% of studies.

Previous research widely reports body weight changes, blood parameters, urinary indices, and bioimpedance analysis as the leading hydration indices [5,222,232]. Acute changes in body weight represent equal changes in hydration status, broadly used to assess hydration [5,222]. This corroborates that “body weight changes” are the most frequent parameter (70%) used in the included studies. In parallel, blood volume and plasma osmolality are the primary variables that are homeostatically regulated [220]. Consistently, both parameters are among the six most found in our review. Even so, the secondary literature argues that blood volume and plasma osmolality are subject to short-term modifications in response to posture change, exercise, food, and fluid intake [220]. Thus, we may infer that neither is a reliable index of hydration status, which raises the question of whether we are looking in the right direction. Or should we seek new routes?

In 1975, Grant et al. [232] segregated the hydration parameters into three main categories: laboratory tests (e.g., serum and plasma osmolality and electrolytes), objective non-invasive measurements (e.g., body mass changes, vital signs), and subjective observations (e.g., thirst) [232]. Due to the elevated variability of personal observations, such a parameter is considered the least reliable measurement [232]. At the same time, the current opinion leans toward urinary parameters, especially urine osmolality and USG, which are the most promising hydration markers [5]. Surprisingly, subjective feelings (36%) were more observed in the included studies than USG (29%) and urine osmolality (18%). Even plasma osmolality (31%), despite constituting an excellent marker to assess acute hydration changes (described as intracellular osmolality), is an invasive method [5] and is more defined than USG and urine osmolality.

Exposure to heat, insufficient fluid ingestion, fever, and physical activity prompt a drop in plasma volume, consequently increasing the osmolality [222]. This dehydration, in turn, raises the arginine vasopressin (AVP) levels and leads to the feeling of thirst [2,6,222]. The results will be a lesser amount of water loss through a decrease in urinary output (because of more significant tubular water reabsorption in the nephron) and a higher consumption of fluid (when available) [222]. The AVP response to dehydration will thus lead to changes in urine color (more concentrated urine characterized by a darker yellowish color), urine osmolality, and USG [222]. Given the intimate link between urine color and USG, some experts recommended that athletes observe their urine color as an index of hydration status [231]. Oppose to the expected, just 9% of the included studies used urine color as a hydration index. That may be because, despite its high specificity, urine color is influenced by certain foods, vitamin supplements, and exercise, which diminishes the reliability of this parameter [5,231].

The available and commonly used hydration parameters have variable accuracy and fall practicality outside controlled environments [233]. However, they are the most suitable for field and large-size sample investigations since they are the most straightforward parameters for sample collection and analysis and the most cost-effective [5,222,232]. Urinary indices should be in the current research spotlight, and we encourage efforts to develop new methods of hydration assessment in future research that are reliable for diverse populations across various settings and, easy to measure, inexpensive, and reproducible to massive cohorts [233]. This strategy will allow us to fill the previously reported gap regarding translation to field “real-world” studies.

The exact impact of dehydration and subsequent weight loss on performance remains unclear. Nonetheless, it is well established that athletes lose water and essential electrolytes like sodium, chloride, and potassium during physical activity through sweating [234]. These electrolytes regulate body water levels and various biological functions such as muscle excitability and cellular permeability [234]. Thus, hydration plays an undebatable key role in sports performance, and it was no surprise that many studies (72%) investigated hydration strategies in performance-based metrics. A total of 22 different outcome measures were reported (Figure 4B), with the most common being rate of perceived exertion (RPE) (55%), VO2max (oxygen consumption) (37%), skill-based performance (36%), time-based performance (31%), and respiratory outputs (e.g., ventilation, respiratory exchange ratio, respiratory quotient) (27.6%) (Figure 4B). The other performance-related outcomes were represented in less than 16% of the studies. Dehydration within the range of 2 to 7% of body mass can adversely affect endurance exercise performance, particularly evident in activities such as cycling time trials [234], one of the most measured outcomes. Specifically, a dehydration level of 3% to 4% may decrease muscle strength by approximately 2%, muscle power by about 3%, and high-intensity endurance by around 10% [235].

Overall, studies focusing only on health outcomes were found to be the least numerous (a noticeable gap), with 46 different parameters listed (Figure 4C); of the 155 studies that included health parameters, 16 outcome measures were included. Heart rate (63%), blood measures (for example, blood lactate, glucose, insulin concentrations) (61%), hematocrit/hemoglobin concentrations (40%), temperature (such as whole-body, rectal, tympanic, skin temperatures) (37%), and subjective feelings (for instance, thirst sensation, stomach fulness gastrointestinal discomfort, mood state) (36%) were the most frequent (Figure 4C). Each of all other health-related outcome measures was observed in less than 13% of studies. Dehydration hampers the body’s ability to regulate heat as sweat production and skin blood flow decrease. With every 1% weight loss due to dehydration, the core temperature can rise by 0.15–0.2 °C, potentially resulting in exertional heat stroke with a core body temperature exceeding 40 °C [220,233]. Dehydration may reduce cardiac stroke volume and increase cardiovascular strain [233]. In addition, dehydration may lead athletes to feel an increment in perceived effort and tiredness, which can, consequently, jeopardize their sports performance and safety [233]. All things considered, it was no surprise that heart rate, temperature, and subjective feelings were among the most observed health outcomes.

## 12. Let Us Not Put All the Eggs into the Same Basket: Future Research Priorities

The representative sample of studies summarized in this review highlights well-delineated trends on (de)hydration research presented and discussed above and outlined in Figure 5.

Observation of Figure 5 allows us to rapidly conclude that hydration is highly explored in recreationally active males more than 18 years old. Cycling under normal environmental conditions is the most used exercise type to study hydration parameters, mainly the rate of perceived exertion, heart rate, blood measures, and body weight changes. The research on this field is still very narrowed to water, with fluids rich in carbohydrates and sodium being the second most common focus of investigation.

Our goal was also to identify research gaps in the existing literature on hydration in healthy individuals and athletes to inform and guide further research and position statements. The current research gaps are outlined in Figure 6. Future research must address such gaps instead of focusing on research topics that are already well investigated.

## 13. Limitations

The evidence presented and discussed in this narrative review is expected to cover a representative sample of (de)hydration strategies in healthy individuals, summarizing data accounting for 4350 individuals from 180 studies (more than 45,000 studies screened). Although we expect that this sample of a high number of studies is representative of research on this topic, as the search performed was not systematic in its nature, there may have been many studies, as well as the grey literature, that were probably left out. Nonetheless, as our goal was to summarize current trends and gaps, the representative sample we collected is considered enough to provide a broad overview of recent research.

## 14. Conclusions

Hydration trends reflect that, despite the vital role played by hydration on athletic performance, it is still an overlooked theme characterized by deficient geographic, sex, age, and level of competition variability. Laboratory sample sizes and simulated environments are also restrictive limitations, underpinning the need to translate scientific laboratory research into the field of “real-world” studies. The inconsistency and lack of clarity in reporting information related to the length of exercise protocol and intake strategy is worrisome and an obstacle to utterly grasping this topic. In contrast, the type of beverages being studied includes a wide range of fluids. Yet, it also displays a new picture of where the hydration horizons should be expanded: hydration with different combinations of fluids.

It was clear from the representative sample of studies that there is no scientific consensus on which parameters best depict hydration status, leading to non-existing global validated hydration indices. Such wide heterogeneity highlights the need to prioritize the search for hydration outcomes that are responsive, easy to measure, inexpensive, accurate, and reproducible to massive cohorts.

## Figures and Tables

**Figure 1 nutrients-16-01709-f001:**
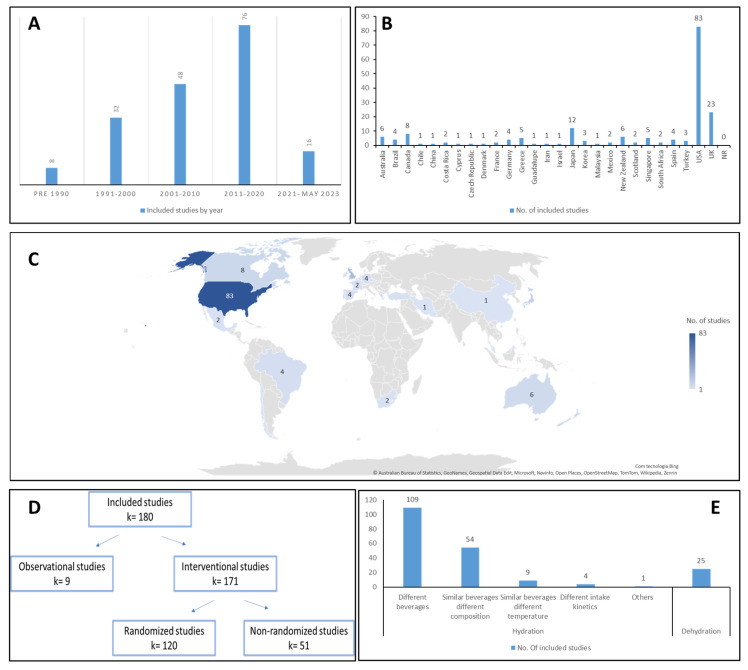
Publication-level distributions: (**A**)—publication dates; (**B**)—country of origin; (**C**)—world map: countries producing the studies; (**D**)—study design; (**E**)—hydration strategy of the articles. Legend: k—number of articles.

**Figure 2 nutrients-16-01709-f002:**
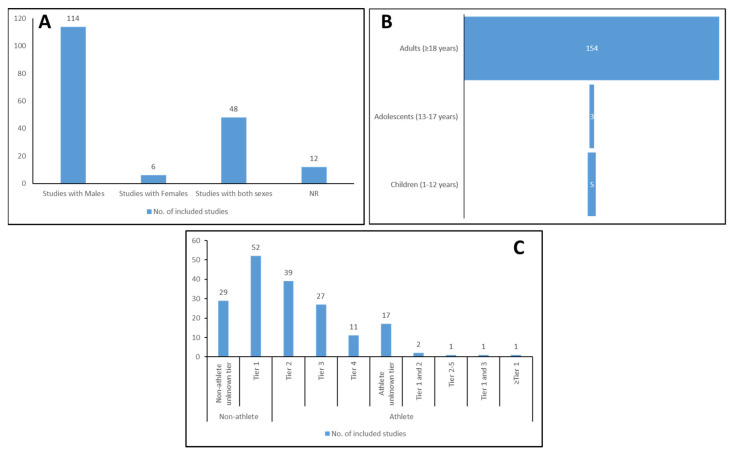
Participants related information: (**A**)—sex; (**B**)—age pyramid; (**C**)—level of competition based on the Participation Classification Framework (PCF) (218). Legend: NR—non-reported; Tier 1—recreationally active; Tier 2—trained/developmental; Tier 3—highly trained/national level; Tier 4—elite/international level.

**Figure 3 nutrients-16-01709-f003:**
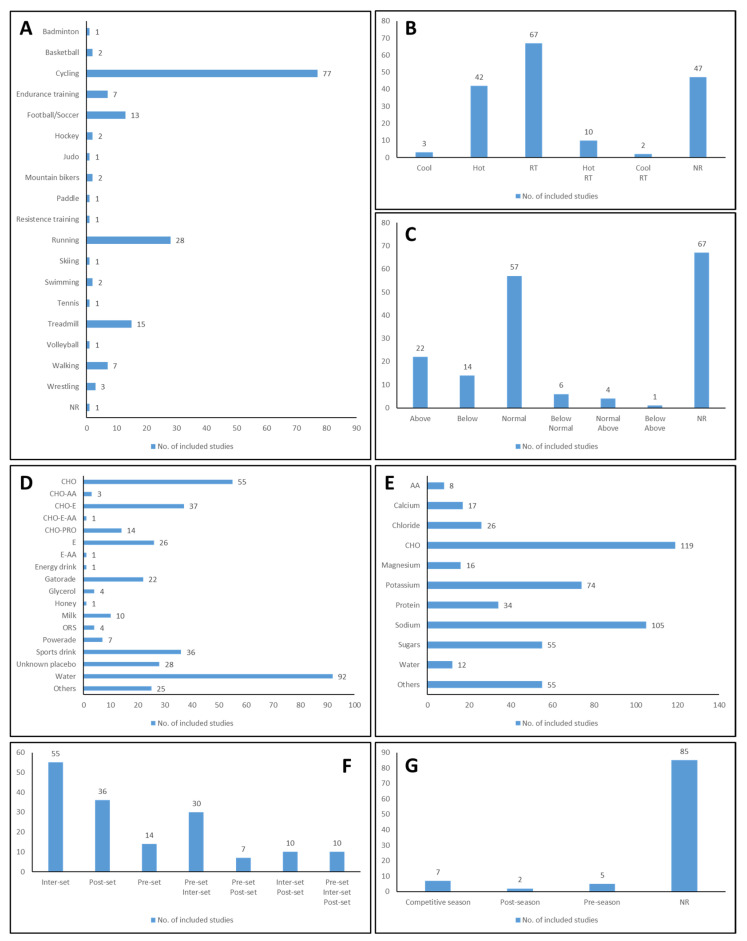
Intervention-related information: (**A**)—exercise types; (**B**)—environmental conditions: temperature; (**C**)—environmental conditions: relative humidity; (**D**)—intake protocol: beverages type; (**E**)—intake protocol: beverages composition; (**F**)—within-session timing; (**G**)—within-season timing. Legend: AA—amino acids, CHO—carbohydrate, E—electrolytes, NR—non-reported, ORS—oral rehydration solution, PRO—proteins, RT—Room temperature.

**Figure 4 nutrients-16-01709-f004:**
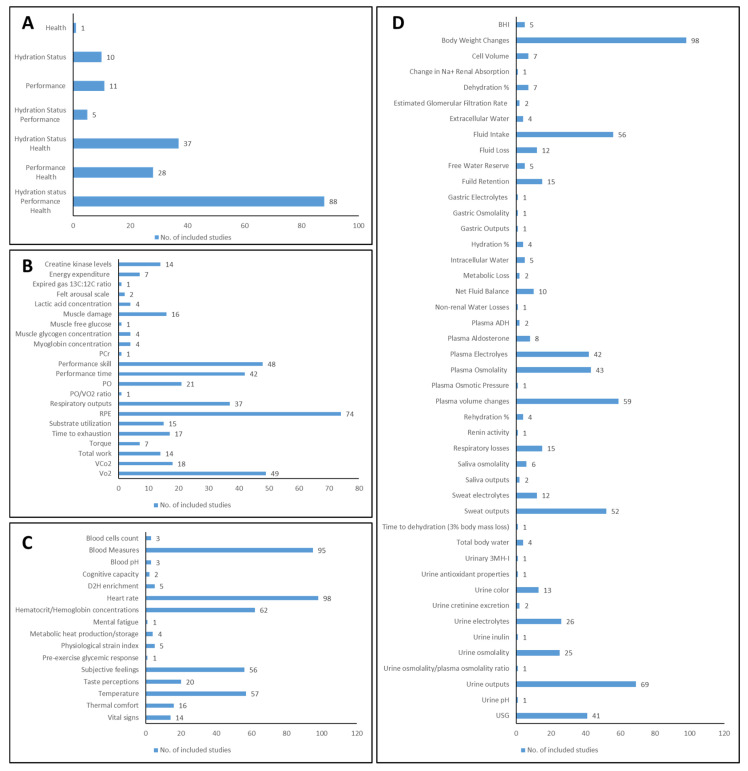
Outcome-related information: (**A**)—outcomes domains; (**B**)—performance outcomes; (**C**)—health outcomes; (**D**)—hydration outcomes. Legend: 3MH-I—3-methylhistidine, ADH—antidiuretic hormone, BHI—beverage hydration index, Hct/hb—hematocrit/hemoglobin concentrations, PCr—phosphocreatine, PO—power output, RPE—rate of perceived exertion, USG—urine specific gravity, VCo2—volume of carbon dioxide breathed out, Vo2—oxygen consumption.

**Figure 5 nutrients-16-01709-f005:**
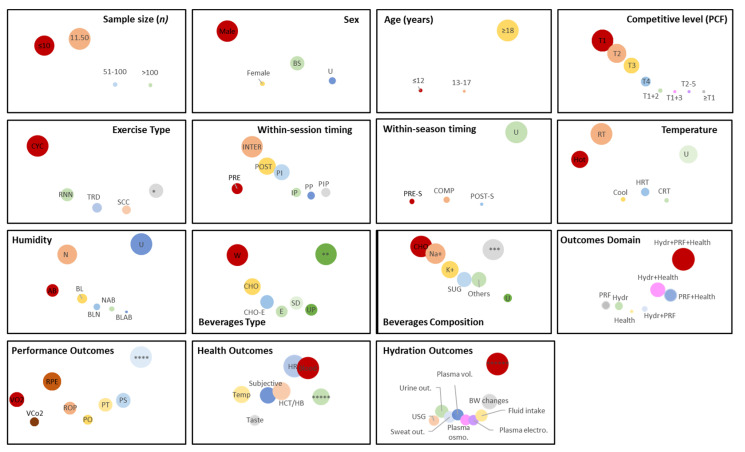
Evidence gap map. Interpretation guide: the area of the circles is proportional within cells but not between cells and is directly related to the proportion of the number of studies for that category. Each trial may be plotted multiple times, as the same trial may fit various categories (e.g., one trial may have analyzed multiple outcomes domains and include multiple exercise types). The number between different categories may differ, as the studies may have provided sufficient information to assess one field but not another. Legend (ordered alphabetically): AB—above, and post-set, BL—below, BLAB—below and above, BLN—below and normal, Blood—blood measures, BS—both sexes, BW changes—body weight changes, CHO—carbohydrates, CHO-E—carbohydrate-electrolytes, COMP—competitive phase, CRT—cool and RT, CYC—cycling, E—electrolytes, HCT/HB—hematocrit and hemoglobin concentrations, HR—heart rate, HRT—hot and RT, Hydr—hydration, INTER—inter-set, Inter-set, IP—inter-set and post-set, K+—potassium, N—normal, Na+—sodium, NAB—normal and above, PCF—performance competitive framework, PI—pre-set and inter-set, PIP—pre-set, Plasma osmo.—plasma osmolality, Plasma vol.—plasma volume, PO—power output, POST—post-set, POST-S—post-season, PP—pre-set and post-set, PRE—pre-set, PRE-S—pre-season, PRF—performance, PS—performance skills, PT—performance time, RNN—running, ROP—respiratory outputs, RPE—rate of perceived exertion, RT—room temperature, SCC—soccer, SD—sports drinks, Subjective—subjective feelings, SUG—sugars, Sweat out.—sweat outputs, T1—Tier 1, T1 + 2—mixed Tiers 1 and 2, T1 + 3—mixed Tiers 1 and 3, T2—Tier 2, T2–5—mixed Tiers from 2 to 5, T3—Tier 3, T4—Tier 4, Taste—taste perceptions, Temp—temperature, TRD—treadmill, U—unreported or unclear, UP—unknown placebo, Urine out.—urine outputs, USG—urine-specific gravity, VCo2, VO2, W—water, ≥T1—at least Tier 1. * Exercise types represented in less than 8% of studies, ** beverage types represented in less than 14% of studies, *** beverages composition represented in less than 19% studies, **** performance outcomes represented in less than 9% of studies, ***** health outcomes represented in less than 9% of studies, ****** hydration outcomes represented in less than 14% of studies.

**Figure 6 nutrients-16-01709-f006:**
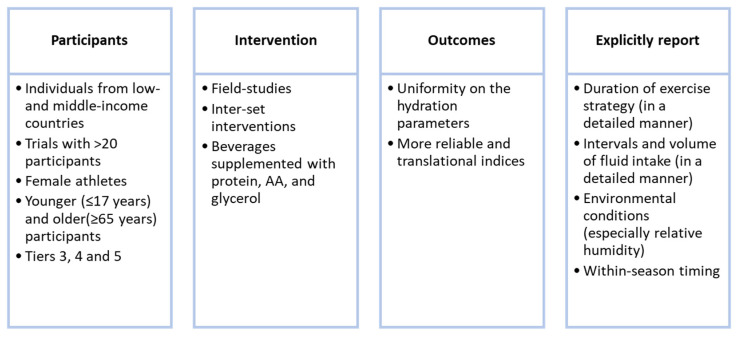
Research priorities regarding hydration in healthy individuals and athletes.

**Table 1 nutrients-16-01709-t001:** Summary of trends on exercise protocol.

Exercise	No. of Articles (k)	Mean	Mode	Range (Min–Max)
Duration (min)	110	112.6	90 (k = 22)	4–930
Sets (rounds of the same exercise)	8	5.5	3 and 5 (k = 2)	3–10
Reps (repetitions within the same round)	6	44	10 (k = 2)	6–200
Kilometers (kms)	24	31.7	20 (k = 3)	3–93
% body weight lost	23	2.3	2 (k = 14)	1.7–4.1
Until exhaustion	29	-	-	-

**Table 2 nutrients-16-01709-t002:** Summary of trends on intake strategy.

Intake Strategy	No. of Articles (k)	Mean	Mode	Range (Min–Max)
Ad libitum	35	-	-	-
% of body weight loss	21	112.3	100 (k = 13)	50–150
% of sweat loss	5	130	150 (k = 3)	100–150
mL of fluid per kg body mass	23	20.8	8 (k = 3)	4–100
% kg body mass	1	2.8	-	-
Concrete fluid volume (mL)	71	2671.9	1000 (k = 15)	275–85,349

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
