# Peer review of "Trends and Missing Links in (De)Hydration Research: A Narrative Review"

_nutrients, 2024, doi:10.3390/nu16111709_

Round 1
Reviewer 1 Report
Comments and Suggestions for Authors
This narrative review was well-summarized with many references cited. The literature about hydration or dehydration were unbalanced in terms of the subjects and conditions. One concern is here.
1. “Over 24 hours, such a balance must equal zero to keep total body water constant (Lines48–49).” Under a basal condition, it is wonder how much is required water intake range, like calculation of basal metabolic rate (at specific temperature, humidity, minimal sweating, and physical and mental state). Out of searched literature, 29% included individuals with recreational activity (Line 176). But basal balance or intake at rest (1 MET) are not known. Unknown reference (the intake or the condition) may make the issue complexed. Did the authors find the information about this? Is it needed to discuss this information?
Minor points
2. Line 114 and the following, and figures and tables. What does “k =” mean?
3. Line 175. The reference for “the Participation Classification Framework” is needed.
4. “Tiers” in Figure 2 are not explained in the caption.
5. Lines 196–199 is a repetition of Lines 139–144. And Line 209 is a repetition of Line 202. This section 5 should be revised.
6. Lines235–236. “5.5 sets and 44 reps” are not understandable. Clarify them.
7. Table 1. The row names, “Sets”, “Reps”, and “Km” are not understandable. It is not self-explanatory. Correct and define them in the footnote.
8. Figure 4A. “Hydration status performance...” What is hidden in “...”?
Author Response
Dear Reviewer,
Thank you very much for taking the time to review this manuscript. We appreciate your insightful comments and suggestions, which have greatly contributed to improving the quality of our work. Please find the detailed responses below, and the corresponding revisions and corrections are highlighted in the re-submitted files using track changes.
Bests Regards.

Reviewer 2 Report
Comments and Suggestions for Authors
The work is interesting, and the graphics included significantly facilitate the understanding of the manuscript. I have minor comments and suggestions.
For keywords, I suggest adding more up to the maximum allowed number. This will increase the article's reach.
Lonia 101 – ‘Studies were published between 1986 and 2023.’ – I understand that the authors analyzed all available literature, correct? Please clarify this.
In the materials I received, there is a visible flowchart, but I do not see any reference to it in the text. Please add this.
I suggest adding the search methodology in the supplementary materials or in the main text. Additionally, why did the authors decide not to conduct a systematic review?
It seems that the graphics use different fonts; please check and standardize this.
L550 – The line spacing is different than in the rest of the text. Please standardize it.
References no. 4 – add page numbers and standardize the title
Author Response
Dear Reviewer,
Thank you very much for taking the time to review this manuscript. We appreciate your insightful comments and suggestions, which have greatly contributed to improving the quality of our work. Please find the detailed responses below, and the corresponding revisions and corrections are highlighted in the re-submitted files using track changes.
Best Regards.
